# Rapid Monitoring of Viable Genetically Modified *Escherichia coli* Using a Cell-Direct Quantitative PCR Method Combined with Propidium Monoazide Treatment

**DOI:** 10.3390/microorganisms11051128

**Published:** 2023-04-26

**Authors:** Yang Qin, Bo Qu, Bumkyu Lee

**Affiliations:** Department of Environment Science & Biotechnology, Jeonju University, Jeonju 55069, Republic of Korea; qinyang2013@jj.ac.kr (Y.Q.); sxqubo@163.com (B.Q.)

**Keywords:** genetically modified microorganisms, detection method, cell-direct, propidium monoazide-quantitative PCR, viable cells

## Abstract

The commercialization of industrial genetically modified microorganisms (GMMs) has highlighted their impact on public health and the environment. Rapid and effective monitoring methods detecting live GMMs are essential to enhance current safety management protocols. This study aims to develop a novel cell-direct quantitative polymerase chain reaction (qPCR) method targeting two antibiotic-resistant genes, *KmR* and *nptII*, conferring resistance against kanamycin and neomycin, along with propidium monoazide, to precisely detect viable *Escherichia coli*. The *E. coli* single-copy taxon-specific gene of D-1-deoxyxylulose 5-phosphate synthase (*dxs*) was used as the internal control. The qPCR assays demonstrated good performance, with dual-plex primer/probe combinations exhibiting specificity, absence of matrix effects, linear dynamic ranges with acceptable amplification efficiencies, and repeatability for DNA, cells, and PMA-treated cells targeting *KmR*/*dxs* and *nptII*/*dxs*. Following the PMA-qPCR assays, the viable cell counts for *KmR*-resistant and *nptII*-resistant *E. coli* strains exhibited a bias% of 24.09% and 0.49%, respectively, which were within the acceptable limit of ±25%, as specified by the European Network of GMO Laboratories. This method successfully established detection limits of 69 and 67 viable genetically modified *E. coli* cells targeting *KmR* and *nptII*, respectively. This provides a feasible monitoring approach as an alternative to DNA processing techniques to detect viable GMMs.

## 1. Introduction

Genetic engineering and biotechnology have been widely applied not only in the creation of genetically modified (GM) crops but also in various industrial fields. In the food and feed industries, genetically modified microorganisms (GMMs) are commonly used to produce additives, enzymes, and condiments through microbiological fermentation [1]. In addition, GMMs are actively researched in the chemical, electronics, energy, resource, and medical industries as industrial genetically modified organisms (GMOs) worldwide. As of 19 December 2022, domestic GMMs have been approved in Korea, 10 of which are intended for industrial applications, with 6 already commercialized for the production of L-alanine, D-psicose-3-epimerase, bioethanol, D-fructose 4-epimerase, L-arabinose isomerase, and 2′-fucosyllactose [2]. However, the rapid development of GMO technologies and the increasing production and industrial use of GMOs have raised concerns regarding their impact on human health and the environment.

Fraiture et al. [1] reported 241 patents related to GM *Bacillus* strains producing bacterial fermentation products. Approximately 59.9% of these patents referred to genes conferring kanamycin resistance, while 33.6, 32.4, and 29.9% referred to genes conferring resistance to ampicillin, bleomycin, and chloramphenicol, respectively. Recent studies have shown the contamination of several commercially available microbial fermentation products with antimicrobial resistance (AMR) genes and live GMMs carrying AMR genes [1,3,4,5]. In Korea, commercialized GMM events commonly use *Corynebacterium glutamicum* and *Escherichia coli* (*E. coli*) as host strains, with the preferred selection markers *nptII* and *KmR* conferring resistance to neomycin and kanamycin, respectively [2]. Despite their widespread use, there are concerns regarding the possible environmental release of GMMs during their commercial production, transportation, and disposal, as they may contain AMR genes. This could facilitate the horizontal transfer of genes to pathogens and other microorganisms, contributing to the spread of AMR.

Therefore, prioritizing safety management is of paramount importance in the development and use of GMOs. A swift and efficient approach for monitoring the release of GMMs is imperative, as there is a risk of potential leaks caused by waste treatment, wastewater treatment, discarded gases during industrial production, and product contamination. Such occurrences can pose a significant hazard. Conventional polymerase chain reaction (PCR), nested PCR, real-time quantitative PCR (qPCR), droplet digital PCR (ddPCR), and next-generation sequencing have been widely used to detect and quantify GMMs [1,4,5,6,7,8,9]. The success of these methods depends primarily on the isolation of DNA. Microorganisms can be present in environmental samples such as contaminated soil, wastewater, and post-treatment products. However, isolating high-quality and high-yield DNA for target microorganisms, particularly in samples with trace contamination by GMMs, can be a daunting and time-consuming task. Additionally, extensive purification may lead to substantial DNA loss and the excessive presence of PCR inhibitors [10]. Although environmental microorganisms have been isolated using culture-based methods, they have several limitations, such as the limited number of cultivable species and prolonged processing time [11]. Heijnen and Medema [12] reported a study on the quantitative detection of *E. coli* and *E. coli* O157 in water samples using a real-time PCR. In their study, they filtered the water sample and incubated it in broth for 20–24 h to isolate the DNA for detection. However, the cell counts observed after incubation may not be the same as the cell numbers present in the original samples. Furthermore, culturing microorganisms only confirms their presence or absence, and does not accurately reflect their abundance. Direct PCR has emerged as a promising molecular technique, which can be applied to samples from various species, including bacteria, fungi, viruses, algae, plants, and humans [13,14,15]. To ensure successful PCR amplification, alkaline lysis pretreatment and high-performance Taq DNA polymerases, rather than DNA isolation, are frequently used. Compared to conventional PCR, qPCR assays require stringent PCR conditions, such as the use of high-quality DNA free from RNA and residual chemicals, optimal concentrations of primers and probes, and the absence of matrix effects, which can induce PCR inhibition. These factors can significantly affect the sensitivity and accuracy of qPCR detection [16], which can limit the performance of direct qPCRs on samples from the environment, plant species, and food industry [15]. For example, humic substances found in soil can act as PCR inhibitors, whereas food matrices may contain proteases and calcium ions, which can inhibit PCRs. Additionally, leaf tissues may contain polysaccharides and phenolic compounds, which can inhibit PCRs [17,18].

This study developed a dual-plex TaqMan-based qPCR method to directly detect and quantify two frequently used AMR genes, *KmR* and *nptII*, in cell cultures. These genes serve as selection markers for domestic, commercial, and industrial GMM events. The direct qPCR approach provides benefits, such as reduced technical requirements and rapid analysis. To avoid false-positive results caused by DNA amplification from dead cells or ejected DNA, propidium monoazide (PMA) treatment was implemented, based on previous studies [19,20]. As a photosensitive dye, PMA selectively penetrates membrane-damaged dead cells and cross-links with their DNA, which prevents PCR amplification. This treatment ensures precision of the obtained results and allows for accurate screening of live GMMs. Therefore, this study primarily aimed to establish safety management protocols for monitoring the environmental emissions of live GMMs used in industrial production using a TaqMan-based qPCR method involving PMA-treated cells.

## 2. Materials and Methods

### 2.1. Bacterial Strains Culture 

Two genetically modified *E. coli* strains, pJ281 and BW25113, were used in this study. The strain pJ281 is derived from the *E. coli* DH5α strain that was modified through the introduction of a plasmid carrying a kanamycin-resistance gene (*KmR*) which encodes the aminoglycoside phosphotransferase APH(3′) type I (Origin: pUC57). The strain BW25113, one of the deletion mutants of *E. coli* K12 named ‘Keio collection’ created by Baba T et al. [21], harbored an *nptII* gene (Origin: pKD13) which encodes Tn5 neomycin phosphotransferase, conferring resistance to antibiotics kanamycin and neomycin, which was inserted into the chromosome regions of *E. coli* K12. A single colony of the two strains was individually inoculated into 10 mL Luria-Bertani (LB) broth (BD DifcoTM, Franklin Lakes, NJ, USA) containing 10 µg/mL kanamycin and grown for 18 h at 37 °C with agitation at 180 rpm.

### 2.2. Measurement of Optical Density and Colony-Forming Units 

Optical density at 600 nm (OD_600_) of the 18 h cell culture was determined by measuring the absorbance in triplicates. Serial cell dilutions from the 18 h cultures were prepared for determining the colony forming units (CFUs) and performing PCR and qPCR analysis of both the strains. A 10^−1^ diluted cell suspension was prepared by adding 100 µL of the 18 h cell culture (cell stock solution, expressed as 10^0^) to 900 µL LB broth. Thereafter, 100 µL of 10^−1^ diluted cell suspension in 900 µL of LB broth resulted in a 10^−2^ diluted cell suspension. Similarly, 10^−3^–10^−8^ diluted cell suspensions were prepared. The cell cultures and diluted samples were stored on ice throughout the experimental process. Each 100 µL of 10^−6^, 10^−7^, and 10^−8^ diluted cell suspensions was spread individually on kanamycin selection plates. CFU/mL was calculated using the following formula:CFU/mL = count number × 1/dilution rate × 1/inoculum,(1)

The average CFU/mL was calculated from three-plate counting of the diluted samples. Ten replications were performed for bacterial cultures and dilution sample preparation to determine the average CFU/mL of the 18 h cell culture (10^0^) for both strains, followed by qPCR and PMA-qPCR assays. 

### 2.3. Primer and Probe Designing for PCR and qPCR Assay

Two primer sets specific for antibiotic resistance genes, *KmR* and *nptII,* were designed to detect the pJ281 and BW25113 strains, respectively. In addition, we designed a primer set targeting the *E. coli* taxon-specific endogenous D-1- deoxyxylulose 5-phosphate synthase gene (*dxs,* GenBank accession number AF035440) as an internal control for PCR and dual-plex qPCR. Three dual-labeled hydrolysis probes, 5′-FAM/3′-BHQ1 for antibiotic resistance genes *KmR* and *nptII*, and 5′-HEX/3′-BHQ1 for *dxs* were designed (Table 1).

### 2.4. DNA Isolation and Quantification

Plasmid DNA of pJ281 and genomic DNA of both the strains were extracted using the Wizard^®^ Plus Minipreps DNA purification system (Promega, Madison, WI, USA) and the Wizard^®^ Genomic DNA purification kit (Promega, Madison, WI, USA), respectively, as per the manufacturer’s guidelines. DNA concentrations were measured using electrophoresis on 1.5% agarose gels and a NanoDrop spectrophotometer (MicroDigital Co., Ltd. Seongnam-si, South Korea), and DNA purity was evaluated using A260/A280 and A260/A230 ratios (values > 1.8).

### 2.5. Generation of DNA Standard Curves

Standard curves of the target genes *KmR*/*dxs* for strain pJ281 and *nptII*/*dxs* for strain BW25113 were respectively created using a ten-fold serial dilution series spanning a range of 1 × 10^3^ to 1 × 10^8^ copies/µL of plasmid DNA and genomic DNA. Genomic DNA sizes were calculated to be 4,534,037 bp for *E. coli* DH5α (pJ281) and 4,639,221 bp for *E. coli* K12 (BW25113), as reported by Song et al. [22] and Engelbrecht et al. [23], respectively. Following the description of Whelan et al. [24], the DNA copy numbers/µL was calculated using the following formula:DNA copy numbers/µL = (6.02 × 10^23^ [copies/mol] × DNA amount [g])/(DNA length [bp] × 660 [g/mol/bp]),(2)

For the strain pJ281, standard curve of *KmR* was generated from serially diluted plasmid DNA by; however, the standard curve of the taxon-specific gene *dxs* was produced from serially diluted genomic DNA by dual-plex. Dual-plex qPCR was performed for another strain BW25113 to simultaneously construct standard curves of the antibiotic gene *nptII* and taxon-specific gene *dxs*. A total of six replications were performed with two qPCR runs to construct a standard curve with the average quantification cycle (Cq) values of each diluted DNA. The resulting Cq values were plotted against the logarithm of the initial template copy numbers. The standard curves generated using the six plotted points, was subjected to linear regression analysis. The slope of each standard curve was used to calculate the qPCR amplification efficiency (E) according to the following equation [25]: Efficiency % = (10^−1/slope^ − 1) × 100,(3)

Calculating the relative standard deviation (RSDr%) and bias% for the two target genes of *KmR* and *dxs* for strain pJ281 and *nptII* and *dxs* for strain BW25113, respectively, allowed for an evaluation of the qPCR’s repeatability and accuracy. Bias% was calculated using the following formula: bias% = (mean DNA copy − true DNA copy)/true DNA copy × 100,(4)

### 2.6. Conventional PCR and Dual-Plex qPCR Analysis

Conventional PCR was performed using a SimpliAmp Thermal cycler (Applied Biosystems, Foster City, CA, USA). Each reaction was carried out in a total volume of 25 µL, comprising 1 µL diluted cell suspension (kept on ice), 10 pmol each of the forward and reverse primers, and 12.5 µL SapphireAmp Fast PCR 2× Master mix (Takara Bio, San Jose, CA, USA). Sterile distilled water (SDW) was used as the non-template control. PCR amplification was performed at 95 °C for 5 min, followed by 30 cycles at 95 °C for 15 s, 60 °C for 10 s, and 72 °C for 15 s. PCR products were analyzed on a 1.5% agarose gel.

Dual-plex qPCR was performed in 48-well plates on a StepOne^TM^ real-time PCR system (Applied Biosystems, Foster City, CA, USA). Amplification reactions contained 10 µL of TOPreal™ qPCR 2X PreMIX (TaqMan Probe for multiplex, Enzynomics, Rebuplic of Korea), 10 pmol of two sets of primers and probes (*KmR*/*dxs* or *nptII*/*dxs*), and 1 µL cell suspension (on ice) or 1 µL DNA template in a final volume of 20 µL. qPCR reactions were performed in triplicates under cycling conditions of 95 °C for 10 min, 40 cycles of 95 °C for 30 s, and 60 °C for 40 s. SDW and LB broth were used as negative controls. Using the cell suspension as a DNA template, a 30 s cell vortex was necessary to homogenously pipette 1 µL. The abundance of target DNA was determined by analyzing the Cq values, which represent the number of cycles needed to reach the default threshold value. The bacterial strains were repeatedly cultured ten times, and after preparing diluted samples as mentioned above, a qPCR analysis was performed in ten intra-runs. The RSDr% was calculated to evaluate qPCR repeatability for the target genes.

### 2.7. PMA Treatment for Viable Cell Quantification

PMA treatment of bacterial cell samples was performed using 20 mM PMA (PMAxx™ Dye, Biotium Inc., Fremont, CA, USA) and exposure to a PMA-Lite™ LED photolysis device (Biotium Inc., Fremont, CA, USA), following the manufacturer’s protocol. Cells (400 µL), including the cell culture (10^0^), serially diluted samples from 10^−1^ to 10^−6^, and LB broth as a negative control were prepared and treated with 1 µL PMA to a final concentration of 50 µM, followed by tube incubation for 10 min in the dark (with shaking at 30 rpm) at 25 °C. After light exposure for 15 min, the sample was placed on ice and kept away from light before use. Dual-plex PMA-qPCR analysis was performed according to the description of qPCR, but primer and probe concentrations were reduced to 5 pmol, and 1 µL PMA-treated cell samples were directly used as DNA templates. To maintain homogeneity of the cell suspension, the sample was vortexed for 15 s before pipetting a 1 µL sample. The qPCR run consisted of two repeats of PMA-treated cell dilutions and one series of non-PMA-treated cell dilutions as controls, and four intra-run PMA-qPCR assays were performed for each bacterial strain. In addition, 12 additional qPCR repeats for 4 PMA-treated cell dilutions from 10^−3^ to 10^−6^ were performed to determine the sensitivity of dual-plex PMA-qPCR to the target gene. The positive reactions of PMA-qPCR were counted, and the limit of detection (LOD_95%_) was analyzed using the Quodata web application [26] available at URL https://quodata.de/content/validation-qualitative-pcr-methods-single-laboratory (accessed on 20 September 2022) in combination with the average viable cell count of the diluted samples. 

### 2.8. Statistical Analyses

Statistical analyses were performed on Microsoft Excel. The average Cq values (mean) and standard deviation (SD) of qPCR and PMA-qPCR were calculated for cell culture, its diluted solution, and plate counting for CFU/mL of both bacterial strains. Correlation analysis was performed to estimate the relationship between OD_600_ and CFU/mL in cell cultures of both strains. In addition, the dynamic range of the linear regression analysis was analyzed for DNA samples, bacterial cell cultures, and PMA-treated cell suspensions, and qPCR efficiency was obtained by tracking the output of the slope, intercept, and R-square.

## 3. Results

### 3.1. Primer and Probe Specificity and Cell-Direct PCR Capability Assessment

The specificity of the primer sets for the two antibiotic-resistant genes, *KmR* and *nptII*, as well as the endogenous taxon-specific gene *dxs*, was tested for PCR amplification. The results confirmed the amplification of a specific fragment using genomic DNA from both bacterial strains, pJ281 and BW25113. No amplification was observed in the negative control (SDW). Furthermore, dual-plex qPCR analysis was conducted for the genome DNA of two strains, which also confirmed the specificity of the primer and probe combinations (Appendix A). To evaluate the cell-direct PCR efficiency of the three primer sets, we PCR-amplified serially diluted samples of cells from the pJ281 and BW25113 strains under identical conditions. The results revealed one specific fragment of the expected size corresponding to 185 bp, 173 bp, and 160 bp for *KmR*, *nptII,* and *dxs*, respectively (Appendix A). The intensity of the PCR products decreased with increasing dilution factor, eventually falling below quantification limits. The detection limits of *KmR* and *dxs* were observed for the dilutions of 10^−5^ and 10^−4^, respectively, in the pJ281 strain. This implies that the pJ281 strain may contain higher copy numbers of the *KmR* gene than those of the endogenous *dxs* gene, which is consistent with the fact that the *KmR* gene was introduced into the strain via plasmid insertion (Appendix A). The detection limits of *nptII* and *dxs* were determined to be 10^−4^ for the BW25113 strain. Thus, the product quantification provided additional confirmation that the *nptII* gene was integrated into the genome of the BW25113 strain and that the copy number of the *nptII* gene was comp0arable to that of the endogenous *dxs* gene (Appendix A). 

### 3.2. Evaluation of Dual-Plex qPCR Performance on DNA

To evaluate the potential for non-specific cross-reactivity in a dual-plex qPCR assay resulting from the use of two sets of primers and probes, we analyzed the qPCR reaction parameters (efficiency, slope, and correlation coefficient) using standard curves derived from genomic and plasmid DNA isolated from the two bacterial strains. When targeting the *KmR* gene, we achieved good performance, with an efficiency of 92.1%, a slope of −3.5268, and a strong linear correlation coefficient of 0.9982 (Figure 1a). This was demonstrated by using six dilution points ranging from 1 × 10^3^ to 1 × 10^8^ DNA copies, which corresponded to plasmid DNA quantities of 8.42 femtogram to 842 picogram (pg) (Appendix A). A standard curve of *dxs* was generated using a dual-plex qPCR assay with *KmR*/*dxs*. Genomic DNA dilutions ranging from 5 pg to 500 ng were used, and the results showed an efficiency of 94.1%, a slope of −3.4728, and a linearity coefficient of 0.9985 (Figure 1b). The repeatability and accuracy of qPCR were assessed using RSDr% and bias%, which were determined based on the Cq values obtained from six reactions. Results showed that the RSDr% values and bias% ranged from 0.35 to 1.39 and −0.61 to 24.12, respectively, both within the acceptable limit of ±25%, as specified in the European Network of GMO Laboratories (ENGLs) guidelines (Appendix A). Thus, the reaction conditions of dual-plex qPCR and the primer/probe combination did not affect the amplification performance of *dxs* when tested in the genomic DNA of the pJ281 strain.

The dual-plex qPCR assay utilizing *nptII*/*dxs* as targets exhibited moderate efficiencies of 94.4% and 90.0%, with corresponding slopes of −3.4636 and −3.5929, respectively (Figure 1c,d). The two standard curves presented suitable linear correlation coefficients (R^2^ > 0.99). These results were obtained using genomic DNA extracted from BW25113 strain over a range of 5–500 ng. The qPCR repeatability and precision results showed that RSDr% values ranged from 0.22 to 1.27, and bias % values ranged from −1.07 to −20.00 (Appendix A), which confirms the suitability of the dual-plex *nptII*/*dxs* primer/probe combinations and ideal qPCR reaction conditions for detecting the BW25113 strain.

### 3.3. Cell-Direct Dual-Plex qPCR Performance 

#### 3.3.1. Evaluation of OD and CFU

After measuring OD_600_ for the 18 h cultured bacterial strains, the diluted cell suspensions were used directly as a DNA template for qPCR and CFU determination. Analysis of the 18 h culture was repeated a total of 10 times, showing an average OD value of 0.76 and 2.39 × 10^9^ of CFU/mL for the pJ281 strain, and 0.78 and 2.62 × 10^9^ CFU/mL for the BW25113 strain. In the pJ281 strain with plasmid insertion, the correlation coefficient of 0.988 between OD and CFU was significant at the 0.01 level (Figure 2a). However, in the BW25113 strain with genome insertion, there was no significant correlation between OD and CFU (Figure 2b). The observed difference in the correlation between OD and CFU in the two bacterial strains may not be attributed to the type of antibiotic gene insertion but rather to their inherent characteristics. Specifically, the BW25113 strain exhibits differences in cell turbidity and tends to form cell clumps, which can interfere with light-scattering measurements. This may limit the use of OD values as a reliable measure of CFU in bacterial cell culture. However, these factors should not affect direct qPCR analysis, as the samples were vortexed to ensure good homogeneity and uniform pipetting of the cell cultures.

#### 3.3.2. Evaluation of Matrix Effect

The potential negative effects on the performance of direct qPCR due to the use of LB broth as the dilution substrate were addressed by conducting a dual-plex qPCR targeting *KmR*/*dxs* and *nptII*/*dxs* on the cell suspension; LB broth and SDW were used as negative controls. There was no significant difference in the Cq values of *KmR*/*dxs* between LB broth (36.59 ± 0.86/36.01 ± 0.52) and SDW (36.59 ± 0.62/35.77 ± 0.27). Similarly, the Cq values of *nptII*/*dxs* were 39.80 ± 0.10/38.81 ± 0.87 for LB broth and 38.82 ± 0.21/38.35 ± 0.40 for SDW, respectively, indicating no significant difference between the two control groups.

#### 3.3.3. Assay Parameters

The qPCR dynamic ranges were evaluated in two bacterial strains, ranging from 10-fold (10^−1^) to 1,000,000-fold (10^−6^) cell dilutions, using 10 intra-qPCR runs. The average Cq values for the four dilution points showed linear regression with amplification efficiencies of 97.6% and 104.8% for *KmR*/*dxs*, and 97.7% and 103.6% for *nptII*/*dxs* (Figure 2c–f). Thus, cell suspensions of the two bacterial strains in this study were suitable for qPCR assays and exhibited reasonable qPCR efficiencies. Additionally, RSDr% values varied from 0.44 to 2.57 for *KmR*/*dxs* and 1.19 to 3.89 for *nptII*/*dxs*, indicating consistent qPCR results using cell suspensions.

### 3.4. Viable Bacterial Cell Detection Using PMA-qPCR

#### 3.4.1. PMA-qPCR Performance for Both Bacterial Strains

For the bacterial strain pJ281, dual-plex qPCR analysis was conducted to detect *KmR* and *dxs* in the stock culture (18 h culture, 10^0^), resulting in average Cq values of 17.49 and 22.61, respectively. Using PMA-treated cell cultures, both targets showed delayed Cq with average values of 18.05 and 22.73, respectively. The dynamic range for diluted cell suspensions treated with PMA, ranging from 10-fold (10^−1^) to 10,000-fold (10^−4^) dilutions, showed a linear regression correlation for both *KmR* and *dxs* as indicated by their respective Cq values. The correlation was expressed as R^2^ and was greater than 0.99 (Figure 2c–e). Upon comparison with the qPCR efficiency of non-PMA-treated cell dilutions, amplification efficiencies of 103.1% and 105.9% for both targets were found to be close to 100%. These findings suggest that the live cells following PMA treatment at each dilution maintained a 10-fold ratio, indicating that the untreated diluted samples contained uniform mixtures of live and dead cells and were evenly diluted for use. In addition, the repeatability of qPCR was assessed; the RSDr% values for each diluted culture ranged from 0.17 to 1.50 for *KmR* and 0.82 to 1.67 for *dxs*. These values did not exceed the limit of ±25% specified in the ENGL (Table 2).

The bacterial strain BW25113 exhibited delayed Cq values for *nptII* and *dxs* in PMA-treated cell cultures (10^0^), with values of 22.16 and 22.72, respectively, compared with non-treated cell cultures with Cq values of 20.37 and 20.53, respectively. Additionally, linearity was observed at five dilution points in the PMA-treated cells, with coefficients of 0.9938 for *nptII* and 0.9909 for *dxs*. Amplification efficiencies of 94.0% and 101.7% were calculated for the two targets (Figure 2d–f). The PMA-Cq values of the cell suspensions showed RSDr % values ranging from 0.09 to 0.66 for *nptII* and 0.34 to 1.47 for *dxs*, which did not exceed the limit of ±25%, as detailed in the ENGL guidelines (Table 2).

#### 3.4.2. Sensitivity of PMA-qPCR to Detect Viable Cells

To determine the sensitivity of the dual-plex PMA-qPCR method with cell suspensions, we conducted 20 qPCR replicates across four diluted points, spanning from 1000-fold (10^−3^) to 1,000,000-fold (10^−6^), with PMA-treated LB broth as the negative control for each qPCR run. Accordingly, we found that the average Cq values for LB broth were 37.69 and 37.86 for the *KmR*/*dxs* combination, whereas the *nptII*/*dxs* combination produced underdetermined results (Table 2). We considered the qPCR runs with Cq values below the negative control as positive runs and used this information to determine the minimum number of viable *E. coli* cells, which could be detected with 95% confidence, which was defined as the limit of detection (LOD). Based on the average CFU, viable cell counts for each dilution were expected to be in the range of 239,000 cells/µL to 2.39 cells/µL from 10-fold (10^−1^) to 1,000,000-fold (10^−6^) dilutions for the pJ281 strain, and 262,000 cells/µL to 2.62 cells/µL for the BW25113 strain. Using the PMA-qPCR method, the LOD values were found to be 10^−5^ and 10^−4^ for *KmR* and *dxs*, respectively, as well as 10^−5^ and 10^−4^ for *nptII* and *dxs*, respectively. These results are consistent with the statistical analysis conducted using the Quodata web application. We determined the LOD at 95% confidence to be approximately 69 viable pJ281 cells with a confidence interval ranging from 29 to 693 and 67 viable BW25113 cells with a confidence interval ranging from 40 to 111 using the dual-plex PMA-qPCR method (Table 3).

#### 3.4.3. Evaluating the Copy Numbers of Antibiotic Genes

We used the taxon-specific gene *dxs*, which exists as a single copy per cell, to calculate the cell count carrying the *KmR* or *nptII* genes based on the Cq values. This strategy allowed consistent determination of the number of CFU. We then evaluated the plasmid copy numbers (antibiotic gene copies) of both bacterial strains by performing dual-plex qPCR with *KmR*/*dxs* and *nptII*/*dxs* primer/probe combinations using DNA, cell suspension, and PMA-treated cell suspension as templates. The results showed that the plasmid copy number of the pJ281 strain was 9.89 copies using DNA, 12.78 copies using cells, and 9.72 copies using PMA-treated cells (Table 4). The average viable cell count using PMA-qPCR was determined by calculating Cq values based on DNA standard curves, and the result revealed approximately 2.97 × 10^9^ copies for *dxs* per mL. When compared to the average CFU/mL of 2.39 × 10^9^ from plate counting for the pJ281 strain, we found a bias% of 24.09, which was within the range of ±25% detailed by the ENGL guidelines, indicating the accuracy and precision of the PMA-qPCR assay. Furthermore, the copy number of the antibiotic gene *nptII* in the BW25113 strain was measured to obtain 0.93 copies using DNA, 0.77 copies using cells, and 1.02 copies using PMA-treated cells (Table 4). The viable cell counts for *nptII* and *dxs* were calculated for 2.68 × 10^9^ copies and 2.63 × 10^9^ copies per mL, with bias% values of −8.63% and 0.49%, respectively. Both values were within the ±25% limit specified in the ENGL guidelines. Consequently, we conclude that the pJ281 bacterial strain carries approximately 9–12 plasmid copies of the *KmR* gene, whereas BW25111 has only one copy of the antibiotic gene *nptII* inserted into the genome region. These findings provide sufficient evidence that the PMA-treated cell-direct dual-plex qPCR technique developed in this study is an effective and feasible approach to monitor viable bacterial cells carrying *KmR* or *nptII* resistance genes.

## 4. Discussion

The present study outlines a dual-plex TaqMan qPCR approach for detecting viable *E. coli* cells carrying the *KmR* or *nptII* genes, which confer resistance to kanamycin in PMA-treated cell suspensions. Two genetically modified *E. coli* strains were selected as the target strains. The kanamycin resistance in these strains differs in that one strain has acquired resistance through the insertion of a plasmid vector, whereas the other strain has resistance conferred by the *nptII* gene inserted into its genome, which provides resistance to both kanamycin and neomycin. To distinguish between genetically modified *E. coli* cells and environmental bacteria with natural kanamycin resistance, we used a taxon-specific endogenous gene, *dxs*, found in *E. coli* as an internal control. We designed three sets of primers targeting *KmR*, *nptII*, and *dxs* genes, and the specificities were verified using in silico analysis according to basic local alignment search tool (BLAST) server at National Center for Biotechnology Information (NCBI), and PCR amplification. In addition, the dual-plex primer and probe combinations demonstrated appropriate efficiencies within the 90–110% limit, as detailed by the ENGL for dual-plex qPCR using purified plasmid DNA and genomic DNA as templates [27]. This suggests that there was no cross-reactivity between *KmR* and *dxs* or *nptII* and *dxs* which could have resulted in dimer formation or other polymerase inhibitions during qPCR assays.

We performed direct qPCR assays using cell suspensions as templates rather than DNA. Cell suspensions not only contain actively growing cells but also the culture medium and various metabolic waste products which accumulate during cell growth. This situation is particularly relevant when detecting microorganisms in environmental samples, such as soil and wastewater, and during food and feed fermentation. These constituents and waste products have the potential to act as polymerase inhibitors, ultimately affecting the effectiveness of qPCR amplification. The ENGL have established acceptance criteria for detecting GMOs in DNA samples, with the slope falling between −3.1 and −3.6, corresponding to amplification efficiencies of 110 to 90%, assuming the absence of PCR inhibitors. However, in the case of overprocessed food/feed samples, the slope of the inhibition curve within −4.1 and −3.1 is acceptable [27]. In this study, cell-direct qPCR amplification efficiencies for *KmR*/*dxs* were 103.1% and 105.9% and those for *nptII*/*dxs* were 94.0% and 101.7%, indicating no significant PCR inhibition caused by the presence of metabolic waste and media in cell culture. Furthermore, both inter- and intra-run cell-direct qPCR showed high levels of repeatability, with all RSDr% values falling within the ±25% range specified by the ENGL. Our study strongly suggested a 30 s vortex before qPCR pipetting to achieve cell suspension homogeneity. Sung and Hawkins [28] conducted a study on the use of cell cultures directly in qPCR assays and noted the potential for reduced assay robustness owing to the presence of cells instead of DNA. Their findings were consistent with those of the present study, as they did not observe any significant differences in the sensitivity or repeatability of the assay to detect mycoplasma in cell culture medium and contaminating DNA. In addition, the OD and CFU values differed among the cultures after 18 h of incubation, and a significant correlation was observed between OD and CFU/mL for the pJ281 strain but not for the BW25113 strain. OD measurements for assessing cell counts are greatly influenced by factors such as the spectrophotometer, wavelength, media type, growth stage, cell morphology, and presence and concentration of secreted compounds [11]. The variations observed between the OD and CFU of BW25113 are more likely to be attributed to its inherent characteristics, such as cell clump formation, rather than the form of antibiotic gene insertion. However, since all cell growth was in the log phase, obtaining consistent qPCR results would not be problematic. 

In this study, we used cell suspensions of 18 h cultures (10^0^) treated with PMA in qPCR analysis, under the same conditions as untreated samples, to detect viable *E. coli* cells. However, we observed significant inhibition of both primer/probe combinations, as evidenced by amplification curves below the baseline and overdelayed Cq values. Sidstedt et al. [10] has elucidated PCR inhibition mechanisms and suggested solutions, such as reducing the concentration of probes and primers, utilizing an inhibitor-tolerant DNA polymerase-buffer system, and employing hydrolysis probes instead of dsDNA-binding dyes for challenging matrices, such as soil and blood. In response, we optimized the PCR by reducing the concentration of primers/probes to 5 pmol for qPCR. This adjustment normalized the amplification curves, suggesting that PMA may act as a PCR inhibitor and reduce the PCR amplification efficiency. Initially, we attempted to prevent PCR inhibition in PMA-treated cell cultures of the pJ281 strain by changing the TaqMan qPCR kit. Two qPCR polymerases, Premix Ex Taq ^TM^ (Takara) and TaqMan ^TM^ Universal PCR Master Mix (Applied Biosystems), were used based on the results reported by Kang et al. [29]; however, PCR amplification efficiency could not be improved. As a result, we designed a new primer/probe combination and confirmed its effectiveness by reducing the concentrations of the primer/probe, as shown in the above results. PMA-qPCR analysis of *KmR*/*dxs* and *nptII*/*dxs* demonstrated linear regressions on diluted cell suspensions, and their efficiencies were within the acceptable range of 90–110%. Furthermore, both RSDr% and bias% were within the ±25% limit, strongly suggesting repeatability and precision of PMA-qPCR in detecting viable *E. coli* cells. These findings suggest that linear regression equations generated from cells and PMA-treated cells can serve as alternative standard curves to DNA.

The cell-direct dual-plex PMA-qPCR assay could detect viable cell counts of 69 and 67 cells for the pJ281 and BW25113 strains, respectively, at 95% confidence levels determined using Quodata analysis. This represents the LOD of the PMA-qPCR assay. The analysis was based on the average CFU/mL, converted into viable cell counts per microliter for qPCR. Therefore, more accurate confidence intervals of 29–693 for pJ281 and 40–111 for BW25113 could be estimated. Based on the plausibility check of LOD 95% for the pJ281 strain, there seems to be a higher average amplification probability at higher dilution levels than that at lower dilution (Appendix A). This result implies several possibilities, such as inhibitory matrix effects, significant variability in qPCR runs, and false positive results. The inhibitor matrix should be excluded because approximately 100% of the PMA-qPCR efficiency of *dxs* indicated that a 10-fold dilution of the sample resulted in a ΔCq of approximately 3.3, but this value could decrease in the presence of inhibitors. All dilution points showed good repeatability. Therefore, we considered the possibility of false positives resulting from sample contamination by *E. coli* with kanamycin resistance from the surrounding environment. To mitigate this risk, all qPCR procedures were conducted on a clean bench. However, the use of PMA-treated LB broth as a negative control is inappropriate because PMA can bind to free DNA. In addition, the Cq values of the pJ281 strain exhibited a slight delay in the internal control of *dxs* after treatment with PMA compared to those of BW25113. Both strains were treated with the same concentration of 50uM PMA, but the difference may be due to a lower proportion of dead cells in the 18h cell culture compared to BW25113. There may be another possibility that the plasmid DNA will be released into the suspension if the cell is dead, but the high number of copies of the plasmid DNA may bind the PMA, resulting in insufficient PMA concentration.

qPCR assays were used to quantify plasmid copy number or antibiotic gene copies following the calculation methods reported by Lee et al. [30]. The copy numbers calculated from the Cq values of DNA, cells, and PMA-treated cells from the two strains were compared. However, because of the possibility of inhibitors in the concentrated samples, we did not employ the Cq values of qPCR from 18h cell culture (cell stock solution, expressed as 10^0^) for copy number calculation. Instead, the average copy number was derived from five diluted samples. The strain pJ281 showed comparable copy numbers between DNA- and PMA-treated cells, suggesting that no plasmid DNA escaped from viable cells during culture. Furthermore, the strain BW25113 was found to have approximately one copy of the *nptII* gene integrated into its genome, as determined in DNA, cells, and PMA-treated cells. In summary, our study highlights the effectiveness of TaqMan-based cell-direct PMA-qPCR assays in accurately monitoring genetically modified *E. coli* cells harboring antibiotic resistance genes, such as *KmR* and *nptII*. By utilizing the linear regression equations of PMA-qPCR generated in this research, we can accurately quantify living kanamycin-resistant *E. coli* cells in a quick and efficient manner. This approach reduces the necessity of complex DNA extraction techniques and saves time while enabling rapid detection of living GMMs that might be released into the environment, including wastewater, air currents, and facility surfaces from large-scale bioreactor industries. However, according to a recent study conducted by Chen et al. [31], identification of GMMs in soil samples was carried out using DNA-based qPCR and PMA-qPCR assays. Due to the presence of large amounts of humic substances, direct qPCR and PMA-qPCR methods may have limitations in accurately detecting GMMs in soil samples, thereby requiring further investigation.

## 5. Conclusions

With the rapid development and commercialization of industrial GMMs, it is important to develop a fast and effective monitoring method to detect living GMM cells in the surrounding environment of bioreactor factories. This is crucial for ensuring safety management and preventing the escape of living GMM cells harboring antimicrobial-resistant genes, which could lead to horizontal gene transfer to other microorganisms and pathogens. In the present study, we developed a cell-direct dual-plex TaqMan-based qPCR method using PMA treatment for directly detecting viable *E. coli* cells resistant to *KmR* and *nptII*. We confirmed the specificity of the primer/probe, assay parameters, and absence of matrix effects for qPCR when using DNA, cells, and PMA-treated cells, respectively. Our results showed acceptable levels of repeatability, precision, and trueness, which fell within the ±25% limit specified in the ENGL guidelines. The cell-direct PMA-qPCR assay allowed an LOD of 69 for *KmR*-resistant and 67 for *nptII*-resistant viable *E. coli* cells. This provides a feasible monitoring approach for detecting living GMMs, which can help evade sophisticated DNA extraction and purification steps. Additionally, this method is time-saving, cost-effective, and minimizes the risk of false positives in environmental samples. 

## Figures and Tables

**Figure 1 microorganisms-11-01128-f001:**
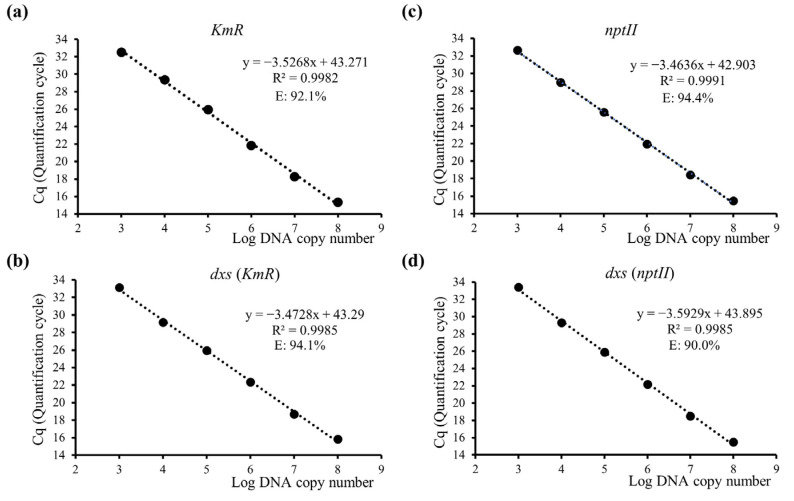
DNA standard curves targeting antibiotic resistant genes of *KmR*, *nptII*, and the *E. coli* taxon-specific endogenous gene, *dxs*, using single-plex or dual-plex quantitative real-time PCR (qPCR) analysis. E—qPCR efficiency; R^2^—linear correlation coefficient. (**a**) The *KmR* standard curve; (**b**) The *dxs* standard curve generated using dual-plex qPCR analysis with the *KmR* combination; (**c**) The *nptII* standard curve; (**d**) The *dxs* standard curve generated using dual-plex qPCR analysis with the *nptII* combination.

**Figure 2 microorganisms-11-01128-f002:**
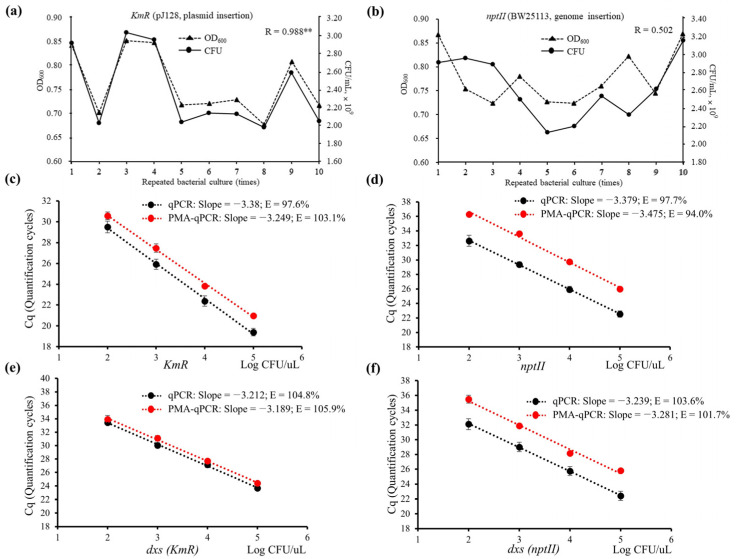
Dual-plex quantitative qPCR and propidium monoazide (PMA)-qPCR assays targeting *KmR*/*dxs* and *nptII*/*dxs* using serially diluted cell suspensions. (**a**,**b**) optical density at 600 nm (OD_600)_ and colony-forming units ((CFU)/mL)) for 10 replicates of *E. coli* strain pJ281 and BW25113 cell culture, respectively; (**c**,**d**) linear regression of qPCR and PMA-qPCR for *KmR* and *nptII*, respectively; (**e**,**f**) linear regression of qPCR and PMA-qPCR for *dxs* of the two strains, respectively. Black triangles with dotted line represent OD_600_ values; black dots with solid line represent CFU/mL. Black and red dots with dotted lines denote average Cq values of qPCR and PMA-qPCR at each serial cell dilution, respectively. E—qPCR efficiency. ** *p* < 0.01.

**Table 1 microorganisms-11-01128-t001:** Two genetically modified *Escherichia coli* strains and primer/probe sequences of target antibiotic resistance genes and their taxon-specific endogenous genes.

Strains	Insertion Form	Target Genes	Primer/Probe Sequence	Size (bp)
pJ281 BW25113	Plasmid	Kanamycin resistance (*KmR*)	*KmR*-F: 5′-CATCATTGGCAACGCTACCTTTG-3′ *KmR*-R: 5′-GAGCCATATTCAACGGGAAACGT-3′ *KmR*-Probe: 5′- FAM-CAACTCTGGCGCATCGGGCTTCCC-BHQ1–3′	185
pJ281 BW25113	Genome	Kanamycin/neomycin resistance (*nptII*)	*nptII*-F: 5′-CGGGTAGCCAACGCTATGTC-3′ *nptII*-R: 5′-GAACTGTTCGCCAGGCTCAA-3′ *nptII*-Probe: 5′-FAM-ACCCAGCCGGCCACAGTCGA-BHQ1–3′	173
*Escherichia coli*	Genome (taxon-specific)	D-1- deoxyxylulose 5-phosphate synthase (*dxs*)	*dxs*-F: 5′-AAGGCATTGTGAAGCGTCGT-3′ *dxs*-R: 5′-CTGGCGGCCATTTCCAGAAT-3′ *dxs*-Probe: 5′-Hex-CGCTGAACGCCACGCTGGTCG-BHQ1–3′	160

**Table 2 microorganisms-11-01128-t002:** PMA-qPCR performance on target antibiotic genes and taxon-specific gene for the two genetically modified *E. coli* strains.

Strains	Target Genes	Dilutions	10^−1^	10^−2^	10^−3^	10^−4^	10^−5^	10^−6^	LB Broth
pJ281	*dxs*	PMA-Cq	24.43 ± 0.20 (Cq:23.67)	27.70 ± 0.24 (Cq:27.17)	31.09 ± 0.39 (Cq:30.04)	33.93 ± 0.57 (Cq:33.42)	37.83 ± 0.07 (Cq:35.97)	37.74 ± 0.81 (Cq:36.45)	37.86 ± 0.32 (Cq:36.26)
		RSDr %	0.82	0.88	1.27	1.67	0.20	2.13	0.47
	*KmR*	PMA-Cq	20.96 ± 0.04 (Cq:19.40)	23.82 ± 0.19 (Cq:22.38)	27.48 ± 0.41 (Cq:25.91)	30.57 ± 0.37 (Cq:29.49)	34.30 ± 0.73 (Cq:32.16)	36.18 ± 0.64 (Cq:34.11)	37.69 ± 0.42 (Cq:36.57)
		RSDr %	0.17	0.78	1.50	1.21	2.14	1.78	1.12
BW25113	*dxs*	PMA-Cq	25.79 ± 0.13 (Cq:22.41)	28.15 ± 0.10 (Cq:25.75)	31.89 ± 0.15 (Cq:29.01)	35.48 ± 0.52 (Cq:32.12)	37.71 ± 0.03 (Cq:35.17)	UD (Cq:37.35)	UD (Cq:38.12)
		RSDr%	0.49	0.34	0.47	1.47	-	-	-
	*nptII*	PMA-Cq	25.99 ± 0.12 (Cq:22.53)	29.76 ± 0.09 (Cq:25.92)	33.61 ± 0.03 (Cq:29.38)	36.29 ± 0.24 (Cq:32.64)	37.84 ± 0.86 (Cq:35.88)	37.06 ± 0.25 (Cq:38.39)	UD (Cq:38.69)
		RSDr %	0.48	0.29	0.09	0.66	2.28	0.68	-

Cq: values detected from cell-direct qPCR, without the use of PMA treatment; PMA-Cq: values detected from cell-direct qPCR after PMA treatment; UD—underdetermined.

**Table 3 microorganisms-11-01128-t003:** Summary of replicate PMA-qPCR runs performed on separate dilution series of suspension cells and limit of detection (LOD) confirmation on viable *E. coli* cells.

Strains	Targets/Dilutions	10^−1^	10^−2^	10^−3^	10^−4^	10^−5^	10^−6^	LOD 95%
pJ281	CFU/µL	239,000	23,900	2390	239	23.9	2.39	69 (29–693)
	dxs	8/8(100%)	8/8(100%)	20/20(100%)	20/20(100%)	16/20(80%)	8/20(40%)	
	KmR	8/8(100%)	8/8(100%)	20/20(100%)	20/20(100%)	20/20(100%)	14/20(70%)	
BW25113	CFU/µL	262,000	26,200	2620	262	26.2	2.62	67 (40–111)
	dxs	8/8(100%)	8/8(100%)	20/20(100%)	20/20(100%)	14/20(70%)	2/20(10%)	
	nptII	8/8(100%)	8/8(100%)	20/20(100%)	20/20(100%)	17/20(85%)	8/20(40%)	

**Table 4 microorganisms-11-01128-t004:** Copy numbers of target antibiotic genes evaluated through qPCR using DNA, cell and PMA-treated cell culture for two genetically modified *E. coli* strains.

Strains	Targets	qPCR (DNA)	Plasmid Copy Number (DNA)	qPCR (Cell)	Plasmid Copy Number (Cell)	PMA-qPCR (Cell)	Viable Cell Count/mL (×10^9^)	Plasmid Copy Number (PMA-Cell)	CFU/mL (×10^9^)	Bias %
**pJ281**	*KmR*	10.64 ± 0.23	9.89	17.49 ± 0.21	12.78	18.05 ± 0.15	28.84 ± 8.67	9.72	2.39 ± 0.03	24.09
	*dxs*	14.61 ± 0.34		22.61 ± 0.21		22.73 ± 0.13	2.97 ± 1.32			
**BW25113**	*nptII*	16.04 ± 0.33	0.93	20.37 ± 0.28	0.77	22.16 ± 0.10	2.68 ± 2.26	1.02	2.62 ± 0.03	−8.63
	*dxs*	15.91 ± 0.36		20.53 ± 0.34		22.77 ± 0.29	2.63 ± 1.39			0.49

Viable cell count/mL: An average cell count calculated by PMA-Cq values according to DNA standard curves; bias%: A ratio of (viable cell count of *dxs*—CFU)/CFU × 100; Plasmid copy number: A ratio of average cell count of the antibiotic gene to that of *dxs*; Cell count: calculated by Cq values according to DNA standard curves.

## Data Availability

The datasets used and/or analyzed within the frame of the present study can be provided by the corresponding author upon a justified request.

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
