# Peer review of "Rapid Monitoring of Viable Genetically Modified Escherichia coli Using a Cell-Direct Quantitative PCR Method Combined with Propidium Monoazide Treatment"

_microorganisms, 2023, doi:10.3390/microorganisms11051128_

Round 1
Reviewer 1 Report
This paper presents a novel cell-direct quantitative PCR method for detecting viable genetically modified microorganisms (GMMs). The method combines PMA treatment and qPCR technology, enabling selective detection of live GMMs. The method offers several advantages, including no requirement for complex DNA extraction techniques, rapid analysis, and avoidance of false-positive results. Additionally, the method can be applied to monitor GMMs that may be released into the environment. The study also tested two antibiotic-resistant genes and demonstrated repeatability and accuracy in DNA, cells, and PMA-treated cells. However, the method was only tested on two antibiotic-resistant genes and requires further validation for other antibiotic-resistant genes.
The manuscript is well-written. The supplementary material is well organized and neat.
It is suggested that the author add a concept map that summarizes the content of the article to increase readability and interest in the manuscript.
The literature in the introductory section is not up to date, please update it with more references to the last two years.
What does (100) in line 493 refer to?
Author Response
We would like to thank you for constructive and insightful comments on the manuscript, which helped us enhance the accessibility of the paper to general audiences and improve the interpretation of our results. We marked modified parts in red in the revised manuscript. We hope that this helps the reviewers to track the improvements in the manuscript.
Point 1: The literature in the introductory section is not up to date, please update it with more references to the last two years.
Response 1: Yes, We added four references. Additional references are as follows:
[7] Daddy Gaoh, S.; Kweon, O.; Lee, Y.-J.; Hussong, D.; Marasa, B.; Ahn, Y. A Propidium Monoazide (PMAxx)-Droplet Digital PCR (ddPCR) for the Detection of Viable Burkholderia cepacia Complex in Nuclease-Free Water and Antiseptics. Microorganisms 2022, 10, 943.
[8] Zhang, J.; Luo, J.; Chen, L.; Ahmed, T.; Alotaibi, S.S.; Wang, Y.; Sun, G.; Li, B.; An, Q. Development of Droplet Digital PCR Assay for Detection of Seed-Borne Burkholderia glumae and B. gladioli Causing Bacterial Panicle Blight Disease of Rice. Microorganisms 2022, 10, 1223.
[9] Pérez-López, J., Alavez, V., Cerritos, R., Andraca-Gómez, G., Fornoni, J., Wegier, A. Residual effects of transgenic cotton on the intestinal microbiota of dysdercus concinnus. Microorganisms 2023, 11, 261.
[31] Chen, L., Li, L., Xie, X., Chai, A., Shi, Y., Fan, T., Xie, J., Li, B. An improved method for quantification of viable fusarium cells in infected soil products by propidium monoazide coupled with real-time PCR. Microorganisms 2022, 10, 1037.
Point 2: What does (100) in line 493 refer to?
Response 2: 100 means cell stock solution, that has cultured for 18 hours. This description can be found at ‘line 113’ in ‘Material and methods 2.2. Measurement of optical density and colony forming units’. To avoid confusion, I mentioned it again in the text (line 502)
Reviewer 2 Report
The reviewed manuscript is dedicated to the design and validation of qPCR-based assay detecting two AMR genes. Here, authors designed duplex quantitative PCR.. The presented results are timely and interesting for scientists, specializing on the field of molecular diagnostics. However, a few comments need to be made and addressed.
Minor issues:
1. More information about similar PCR-based tests in Introduction and Discussion sections would greatly increase the readability of the manuscript.
2. The used genes can contain polymorphic sites which can be under primers and probes. Thus, the PCR efficiency can be reduced leading to decreased analytical sensitivity. Thus, more information needs to be provided about primers design and KmR, nptII, and dxs genes conservativity.
3. Authors are encouraged to clarify, how true DNA amount was determined, especially, for the KmR plasmid, which copy number in cells can vary.
Minor issues:
1. Page 2, line 77: “matrix” — plausibly, “template” was assumed.
2. Page 3, lines 99-101: “Strain pJ281 was transformed with a plasmid carrying a kanamycin-resistance gene (KmR) encoding aminoglycoside phosphotransferase APH(3') type I (Origin: pUC57) into E. coli DH5α cells.” — was the pJ281 strain originated from the DH5α strain transformed by a plasmid with KmR? If so, please, reformulate the sentence.
3. Table 1 — authors are encouraged to clarify why amplicons longer than 150 bp were used in the study. Commonly, the less length an amplicon has, the more is its amplification efficiency.
4. Page 12, line 485: “50uM PMA”
5. Page 13, line 507: “humic metrics”
Author Response
We would like to thank you for constructive and insightful comments on the manuscript, which helped us enhance the accessibility of the paper to general audiences and improve the interpretation of our results. We marked modified parts in red in the revised manuscript. We hope that this helps the reviewers to track the improvements in the manuscript.
Point 1: More information about similar PCR-based tests in Introduction and Discussion sections would greatly increase the readability of the manuscript.
Response 1: Yes, several references and sentences were discussed in Introduction and Discussion sections as follows:
Introduction section: We added a study report of Heijnen and Medema (2006) (line 69-73)
Discussion section: We added a additional study of Chen et al. (2022) (line 515-519)
We also additionally cited four references [7], [8], [9], [31].
Point 2: The used genes can contain polymorphic sites which can be under primers and probes. Thus, the PCR efficiency can be reduced leading to decreased analytical sensitivity. Thus, more information needs to be provided about primers design and KmR, nptII, and dxs genes conservativity.
Response 2: We design primers and probes using 'Oligo Architect TM online' tool and blast qPCR amplicons against NCBI database. The amplicons of antibiotic resistant genes KmR and nptII showed 37 and 16 hits in microbe database respectively. There were not any hit on the target E. coli strains (DH5a and K12), which used for genetically modified microorganisms. Therefore, the polymorphic sites should not affect the analytical sensitivity. Also, the amplicon of dxs showed most hits in E. coli strains. Although we can find several hits with 79.14%-99.36% identity in several microbes such as Shigella boydii, Citrobacter pasteurii, Raoultella electrica, Raoultella and so on, these microbes exists in natural without KmR or nptII resistant genes. By using dual-plex qPCR, only AMR resistant E. coli cells can be detected.
Point 3: Authors are encouraged to clarify, how true DNA amount was determined, especially, for the KmR plasmid, which copy number in cells can vary.
Response 3: For KmR resistant E. coli strain pJ281, we used genome DNA as template for dual-plex qPCR with KmR/dxs primer/probe combinations. The genome DNA concentration is determined to be 500 ng. However, how many plasmid DNA contained in genome DNA samples can not be accurately determined. Therefore, qPCR assays using DNA samples, cell samples and PMA-treated cell samples were performed to determine approximately 9–12 plasmid copies of the KmR gene in pJ281 strain.
Minor issues:
Point 4: Page 2, line 77: “matrix” — plausibly, “template” was assumed.
Response 4: We understand the 'matrix' to refer to certain types of substances or components. It may be present in DNA samples or PCR reactions including RNA or proteins that have mixed with the DNA sample, various chemicals or metabolites that are not intended for use in PCR reactions, and so forth.
Point 5: Page 3, lines 99-101: “Strain pJ281 was transformed with a plasmid carrying a kanamycin-resistance gene (KmR) encoding aminoglycoside phosphotransferase APH (3') type I (Origin: pUC57) into E. coli DH5α cells.” — was the pJ281 strain originated from the DH5α strain transformed by a plasmid with KmR? If so, please, reformulate the sentence.
Response 5: According to reviewer’s suggestion, We reformed the sentence as follows: (line 105-107) The strain pJ281 is derived from the E. coli DH5α strain that was modified through the introduction of a plasmid carrying a kanamycin-resistance gene (KmR) which encodes the aminoglycoside phosphotransferase APH(3') type I (Origin: pUC57).
Point 6: Table 1 — authors are encouraged to clarify why amplicons longer than 150 bp were used in the study. Commonly, the less length an amplicon has, the more is its amplification efficiency.
Response 6: To optimize qPCR, it is generally recommended to use PCR amplicons that are between 80 and 200 base pairs in size. In order to increase specificity, we chose the target PCR size of 100-200 base pairs by utilizing the primer and probe design tool 'OligoArchitect TM online', which can be found at http://www.oligoarchitect.com/ShowToolServlet?TYPE=DPROBE. The predicted qPCR amplicons for the target genes were then blasted against the NCBI database to ensure their specificity. We prioritized the specificity of the primer and probe sets relative to the amplicon size in order to minimize the possibility of false positives.
Point 7: Page 12, line 485: “50uM PMA”
Response 7: The PMA concentration may not be suitable for all experimental conditions and should be evaluated on a case-by-case basis. Therefore we did not mentioned the exact concentration.
Point 8: Page 13, line 507: “humic metrics”
Response 8: Yes, it is an inaccurate description. ‘Humic metrics’ is modified to ‘humic substances’ (line 518)
Reviewer 3 Report
This publication does a fair job of illustrating the usefulness of direct qPCR in identifying antibiotic-marked-GMM strains. Addressing appropriate methodology for GMM strains likely to escape biotech facilities is a critical one for biosafety monitoring. This study serves as a helpful reminder to all geneticists to consider population impacts in the background controls and, presumably, the mutant lines under consideration.
The paper is well-written, comprehensive in methodology issues, and adds a novel perspective to the field. The IMRAD structure is well-organized and given in a logical sequence. The experimental dataset is adequate for answering research questions and explaining the relevance of observations..
Author Response
We would like to thank you for constructive and insightful comments on the manuscript, which helped us enhance the accessibility of the paper to general audiences and improve the interpretation of our results. By the other reviewer's commends, we marked modified parts in red in the revised manuscript. We hope that this helps the reviewers to track the improvements in the manuscript.